# Cognitive Underperformance in a Mixed Neuropsychiatric Sample at Diagnostic Evaluation of Adult ADHD

**DOI:** 10.3390/jcm12216926

**Published:** 2023-11-04

**Authors:** Hui Dong, Janneke Koerts, Gerdina H. M. Pijnenborg, Norbert Scherbaum, Bernhard W. Müller, Anselm B. M. Fuermaier

**Affiliations:** 1Department of Clinical and Developmental Neuropsychology, Faculty of Behavioral and Social Sciences, University of Groningen, 9712 TS Groningen, The Netherlands; h.dong@rug.nl (H.D.); janneke.koerts@rug.nl (J.K.); g.h.m.pijnenborg@rug.nl (G.H.M.P.); 2LVR University Hospital, Department of Psychiatry and Psychotherapy, Faculty of Medicine, University of Duisburg-Essen, 45147 Essen, Germany; norbert.scherbaum@uni-due.de (N.S.); bernhard.mueller@uni-due.de (B.W.M.); 3Department of Psychology, University of Wuppertal, 42119 Wuppertal, Germany

**Keywords:** performance validity, symptom validity, embedded validity testing, cognitive underperformance, neuropsychological assessment, adult ADHD

## Abstract

(1) Background: The clinical assessment of attention-deficit/hyperactivity disorder (ADHD) in adulthood is known to show non-trivial base rates of noncredible performance and requires thorough validity assessment. (2) Objectives: The present study estimated base rates of noncredible performance in clinical evaluations of adult ADHD on one or more of 17 embedded validity indicators (EVIs). This study further examines the effect of the order of test administration on EVI failure rates, the association between cognitive underperformance and symptom overreporting, and the prediction of cognitive underperformance by clinical information. (3) Methods: A mixed neuropsychiatric sample (N = 464, ADHD = 227) completed a comprehensive neuropsychological assessment battery on the Vienna Test System (VTS; CFADHD). Test performance allows the computation of 17 embedded performance validity indicators (PVTs) derived from eight different neuropsychological tests. Further, all participants completed several self- and other-report symptom rating scales assessing depressive symptoms and cognitive functioning. The Conners’ Adult ADHD Rating Scale and the Beck Depression Inventory-II were administered to derive embedded symptom validity measures (SVTs). (4) Results and conclusion: Noncredible performance occurs in a sizeable proportion of about 10% up to 30% of individuals throughout the entire battery. Tests for attention and concentration appear to be the most adequate and sensitive for detecting underperformance. Cognitive underperformance represents a coherent construct and seems dissociable from symptom overreporting. These results emphasize the importance of performing multiple PVTs, at different time points, and promote more accurate calculation of the positive and negative predictive values of a given validity measure for noncredible performance during clinical assessments. Future studies should further examine whether and how the present results stand in other clinical populations, by implementing rigorous reference standards of noncredible performance, characterizing those failing PVT assessments, and differentiating between underlying motivations.

## 1. Introduction

### 1.1. Noncredible Symptom Report and Test Performance in Adult ADHD: Concept, Reasons, Consequences

A large body of evidence from numerous research studies demonstrates that the clinical evaluation of attention-deficit/hyperactivity disorder (ADHD) in adulthood must consider the possibility of noncredible reports on self-report measures and noncredible performance on neuropsychological tests. The reasons for noncredible symptom reports or performance can be manifold between and within individuals, including conscious and unconscious forms, and often depend on the heterogeneous composition of the sample, their motivation, and the context of the assessment [1,2,3,4]. While the underlying reason for noncredible symptom reports and performance may be difficult to determine on the individual level, a non-trivial proportion may be motivated to deliberately feign ADHD in order to obtain access to external or internal benefits. Such benefits may span a broad range, such as receiving extra time for exams or assignments in college or on high-stakes examinations, special accommodation or bursaries provided by the university or government, access to stimulant medications, or seeking an excuse for (academic) failure or unreliable behavior in social situations [1,5,6,7,8]. Noncredible symptom reports and test performance in ADHD assessments have many unfavorable consequences, including distorting diagnostic assessment and treatment plans, spanning pharmacological but also nonpharmacological interventions (e.g., see [9,10,11]), imposing substantial costs to society for illegitimate and unnecessary assessments and treatments, increasing the risk of adverse health effects due to excessive treatment, unjustified use of limited medical resources and undermining public confidence in clinical diagnostics and treatment, which may also contribute to stigmatizing beliefs towards ADHD [12].

### 1.2. Embedded versus Stand-Alone Validity Indicators

Thus, there is a strong need for well-validated measures to assess the validity of symptom reports (symptom validity tests (SVTs)) and performance on cognitive tests (performance validity tests (PVTs)). Specifically, symptom validity describes the degree to which an individual’s symptomatic complaint on self-report measures is reflective of the true experience of symptoms, including self- and observer-reported rating scales, whereas performance validity describes the degree to which a person’s test performance is reflective of true cognitive ability, including personality inventories and routine neuropsychological tests [13]. Embedded validity indices (EVIs) of SVTs and PVTs are measures embedded in or derived from clinical instruments that are used in routine clinical practice to assess functioning or symptomatology. Compared to stand-alone PVTs, the use of EVIs derived from cognitive tests is attractive because they do not require additional test-taking time, may not be susceptible to coaching, and cover various clinical constructs and/or domains in validity assessment during routine clinical assessment [14]. In particular, the use of multiple embedded PVTs enables continuous validity assessment on neuropsychological batteries to account for potential fluctuations in the examinees’ effort and motivation that may occur for a variety of reasons during the assessment process [15]. Stand-alone PVTs, in contrast, increase the length of the clinical assessment by requiring additional administration time. Furthermore, stand-alone PVTs are still limited in the range of functions they assess, as the majority of well-validated stand-alone PVTs are based on memory functions, such as the Word Memory Test [16], the Medical Symptom Validity Test [17], or the Test of Memory Malingering [18], which has restricted values in the assessment of attention disorders [19]. However, a serious disadvantage of EVIs is that they are by nature more sensitive to genuine cognitive impairment and, therefore, need to be thoroughly validated to prevent the confusion of genuine cognitive impairments with noncredible test performance (see invalid-before-impaired paradox, [20]). In this respect, it has been suggested that this disadvantage can be overcome by aggregating multiple performance EVIs [21].

### 1.3. Base Rates of Noncredible Performance in the Clinical Evaluation of Adult ADHD

Base rate estimations of non-credible performance of adults during an ADHD assessment made use of different and diverse measures of performance validity, including one or several embedded or stand-alone PVTs or a combination of those. Based on the liberal criterion of one single PVT to determine invalid performance, research reported base rates ranging from 15 up to 49% of individuals at clinical evaluation of adult ADHD showing indication of noncredible test performance [22,23,24,25]. More recent studies using stricter criteria, i.e., positive results on multiple independent PVTs to determine noncredible performance, reported base rates ranging from 9% to 19%. Among those, Ovsiew et al. [26] reported a base rate of 16% of invalid cognitive performance in 392 adults undergoing ADHD assessment when using the criterion of ≥2 PVT failures. Based on the same criterion of determining invalid cognitive performance (≥2 PVT failures), Mascarenhas et al. [27] found a base rate of 9.4% of invalid performance in a large sample (*n* = 1045) of post-secondary students who underwent a comprehensive psychoeducational evaluation. Hirsch et al. [3,28] reported failure rates on one PVT (Amsterdam Short Term Memory Test, ASTM) between 32.1 and 49.3% on samples of 196 and 700 adults with ADHD, respectively. When applying a stricter criterion for noncredible performance of positive results on two PVTs on the same samples, base rates dropped to 18.9–27.3% [3,28]. Hence, differences in PVT methodology, rigorousness of determining invalid cognitive performance, and sample composition across studies are, obviously, major factors for the heterogeneity of base rate estimations. Determining accurate base rates of noncredible performance is relevant, as it allows clinicians and researchers to calculate positive and negative predictive values for a given validity measure with greater confidence, thus preventing confounding interpretation of clinical data and providing essential psychometric information for interpreting sensitivity and specificity contextually [29].

### 1.4. Relationship between Symptom and Performance Validity

Current guidelines recommend that a thorough credibility assessment of a neuropsychological evaluation should utilize both SVTs and PVTs, as they seem to provide largely unique information regarding the credibility of symptom responses and test performance [30,31]. SVTs as indicators of symptom overreporting and PVTs as indicators of underperformance seem to represent distinct constructs, as concluded from studies utilizing factor analytic techniques [13,32,33]. For example, Van Dyke et al. [13] supported a 3-factor model: cognitive performance, performance validity, and symptom self-report (with symptom validity measures loading on the last factor), indicating that PVTs and SVTs loaded on different factors and may carry unique information. However, some restriction to this notion is given by several studies showing significant (albeit weak to moderate) association between SVTs and PVTs [34,35,36] and insufficient evidence for samples of mixed neuropsychiatric patients, which warrants further examination.

### 1.5. ADHD Research Using Embedded Validity Indicators

Efficient and universally applicable EVIs in ADHD assessment should be derived from cognitive tests that are routinely applied in the neuropsychological assessment of adults with ADHD across settings. Because of the commonly observed impairments in several aspects of attention and executive control [37,38,39], clinicians and researchers recommend tasks for sustained attention, distractibility, and inhibitory control to be among the three most relevant functions for routine cognitive assessment [40]. The widespread popularity and common availability in neuropsychological practice, including the assessment of adults with ADHD, is recognized for continuous performance tests (CPT; [40,41,42]). CPTs were designed to assess one or several aspects of sustained attention and vigilance and became growingly popular to serve as EVIs across populations and assessment settings [43,44,45,46]. Although variants of CPTs vary widely in task characteristics and stimulus material, the accuracy (expressed in errors of omissions and commissions) and speed of responses (expressed in reaction times and variability of reaction times) are the most commonly derived test variables and provide solid predictive accuracy for noncredible cognitive performance [47]. Other cognitive tests that are commonly applied in the routine neuropsychological assessment of adult ADHD, and that also hold promising features for credibility testing in ADHD assessments, include the Wechsler Adult Intelligence Scale-Fourth Edition (WAIS-IV) Digit Span (testing short-term and working memory, with a sensitivity of 35–48% and specificity ≥ 85%; [48]), the Trail Making Test (test for processing speed and mental flexibility, with a sensitivity of 29–36% and specificity ≥ 89%; [49]), and verbal fluency (testing divergent thinking, with a sensitivity of 29% and specificity of 91%; [49]). However, the mentioned studies did not examine the utility of a large number of EVIs across multiple cognitive domains on the same sample of participants, which restricts a direct comparison of their value as EVIs.

### 1.6. Introduction of Recently Developed EVIs by Becke et al., 2023

Based on the current state of evidence, only a few studies are available that help research and practice determine which EVIs are most helpful in the diagnostic evaluation of ADHD. Recently, Becke et al. [50] reported findings from an analogue study on 247 individuals, of which 57 were genuine patients with ADHD and 151 typically developing individuals who were instructed to perform the assessment as if they had ADHD. All participants were requested to complete a comprehensive neuropsychological battery of eight neuropsychological tests that offer a broad range of 21 test variables (i.e., Cognitive Functions Adult ADHD; CFADHD; [50,51]). With the claim of reaching specificity levels of at least 90% for all measures, sensitivity rates ranged from 0 to 65% per test variable. The authors further concluded from their data that tests of selective attention, vigilance, and inhibition are likely most useful in detecting noncredible performance in an ADHD assessment. While this simulation study provides promising results with potential value for practice, further validation in the clinical setting is warranted.

The present study employs 17 of the recently developed EVIs on eight different neuropsychological tests derived from the CFADHD [50,52] on a mixed sample of 464 individuals consecutively presenting for the clinical evaluation of adult ADHD. The first aim of this study was to establish base rates of noncredible performance per function and test (variable). In estimating base rates, this study takes into consideration current guidelines and practice standards of determining noncredible cognitive performance by positive results on at least two independent validity measures [31,53,54]. Second, this study is the first that aims to explore how indicators of noncredible cognitive functions emerge along the performance of an extensive battery of about two hours of administration time and how they relate to each other. Third, measures of performance validity will be associated with measures of symptom validity to provide further evidence of their differentiation. Finally, and fourth, characteristics of those showing increasing evidence for noncredible cognitive performance will be explored in a multivariate prediction model based on a range of routine clinical measures, including symptom self- and other-reports, the discrepancy between informants’ evaluations, self-reported level of cognitive functioning, and other indicators of psychopathology. Such a characterization may help to define groups of people who are more likely to fail performance validity testing and, thus, may support efficient clinical assessment to differentiate between invalid and valid clinical data. Based on prior research and the objectives of the current study, we formulate the following research hypotheses: (1) Tests for attention may show the highest base rates of noncredible performance, (2) EVI failure rates across the order of test administration do not show any seemingly relevant effect of time on task but are all closely associated with each other, (3) Although there may be some overlap between SVTs and PVTs, the two constructs are dissociable, and (4) Clinical routine measures are not useful to predict cognitive underperformance.

## 2. Materials and Methods

### 2.1. Procedure and Participants

#### 2.1.1. Recruitment and Assessment

All participants were recruited from the ADHD outpatient clinic of the University of Duisburg-Essen i.e., the Department of Psychiatry and Psychotherapy, LVR Hospital Essen, Germany from 2017 to 2021. Participants were referred for a comprehensive diagnostic assessment of ADHD because of ADHD suspected by local neurologists, psychiatrists, GPs, or by themselves. The diagnostic assessment was based on the criteria as outlined in the Diagnostic and Statistical Manual of Mental Disorders, 5th Edition (DSM-5; [55]) and current empirically informed guidelines [56]. Adult first-time diagnostic criteria for ADHD were employed because information on participants’ formal childhood ADHD diagnoses could not be reliably retrieved for all individuals. The diagnostic assessment procedure included semi-structured interviews evaluating ADHD and related psychopathology, including the Wender–Reimherr Interview [57] and the Essen-Interview-for-schooldays-related-biography [58]. Further, self- and observer-completed questionnaires on current and retrospective ADHD symptoms in childhood were administered in order to quantify the presence and severity of experienced symptoms. The diagnostic evaluation also included the assessment of impairments in major areas of individuals’ lives, which was supported by objective indications such as evidence derived from school reports and reports of failure in academic and/or occupational achievement, and comprised multiple informants wherever possible (e.g., employer evaluation, partner or parent report). No exclusion criteria were applied in order to obtain a naturalistic sample of individuals presenting in an ADHD outpatient referral context with a broad spectrum of characteristics. The breadth of characteristics is considered important for our research goals in estimating realistic base rates and representative prediction models.

Questionnaires had to be completed at home prior to the appointment at the outpatient clinic. The diagnostic evaluation was followed by a neuropsychological assessment using cognitive tests on the Vienna Test System [51] on the same or another day of convenience for the examinee. The test assessment is known to be no diagnostic source for the clinical diagnosis of ADHD; however, the results of the cognitive assessment were accessible to clinicians and may have guided clinical decision-making. Cognitive testing took about two hours and was led by a trained psychologist or neuropsychological test assistant under close supervision. The test order of the assessment is displayed in Figure 1 from left to right. Both diagnosis and neuropsychological assessments are part of the standard clinical routine for all participants referred to the ADHD outpatient clinic of the University of Duisburg-Essen, Germany. All individuals provided written informed consent and agreed that the data collected in their routine clinical assessment may be used for academic research. Participation in the study was voluntary and unpaid, and it was emphasized that consent to participate in the study would not affect their clinical assessment or treatment. However, the purpose of this specific study was not presented to the participants. This was a retrospective study with no a priori research plan and ethics protocol. Ethical permission was asked for towards the end of data collection. This procedure was approved by the Ethical Review Board of the Faculty of Medicine, University of Duisburg-Essen, Germany, with an approval date of 18 June 2020, approval number 20-9380-BO.

#### 2.1.2. Characteristics of Participants

A total of 464 individuals were included in this study, of which 227 (48.9%) participants were diagnosed with ADHD after a comprehensive assessment and 237 (51.1%) participants were not. Further, 196 (42.2%) individuals of the total sample met diagnostic criteria for one or more psychiatric disorders other than ADHD (77 of those comorbid with ADHD). Diagnostic assessments further showed that 118 (25.4%) individuals did not meet the diagnostic criteria for any psychiatric disorder. In the ADHD group (*n* = 227), 175 individuals were diagnosed with the combined symptom presentation, 26 individuals with the predominantly inattentive presentation, and 2 individuals with the predominantly hyperactive/impulsive symptom presentation. For 24 individuals with ADHD, the symptom presentation was not reported. Psychiatric conditions other than ADHD (in the entire sample) spanned a broad range of diagnostic categories, including mood disorders (*n* = 121), addiction disorders (*n* = 42), personality disorders (*n* = 28), anxiety disorders (*n* = 18), adjustment disorders (*n* = 15), obsessive compulsive disorders (*n* = 6), post-traumatic stress disorders (*n* = 5), schizophrenia (*n* = 4), eating disorders (*n* = 4), intellectual development disorders (*n* = 3), autism disorders (*n* = 1), somatoform disorders (*n* = 2), and others (*n* = 12). Educational level was assessed on a scale of 5 levels, with no school-leaving qualification (*n* = 10), compulsory schooling or secondary school completed (*n* = 90), completed technical school or vocational training (*n* = 144), higher school with university entrance qualification (*n* = 139), and university or college degree (*n* = 79). Education level was not reported in two cases. The characteristics of all participants are presented in Table 1.

Data of participants included in the present study overlapped with smaller data chunks used in previous research on different research questions (e.g., see [59,60,61]). Whereas previous topics of research included the diagnostic and clinical value of various clinical measures, the present study could be distinguished by its focus on symptom and performance validity. The measures and their application as EVIs are described below.

### 2.2. Materials

#### 2.2.1. Self- and Other-Report Symptom Questionnaires

The long version of the Conners’ Adult ADHD Rating Scales (CAARS; [62,63]) was used to assess the presence and the severity of current ADHD symptoms. The CAARS consists of 66 items, each rated on a 4-point Likert scale ranging from 0 (not at all/never) to 3 (very much/very frequently). Sum scores can be calculated for eight subscales. The present study reports and makes use of the ADHD Index from both the self-report (CAARS-SR) and observer report (CAARS-OR). The ADHD Index refers to the items that best distinguish individuals with ADHD from non-clinical individuals. In addition to raw scores of the CAARS ADHD Index self- and observer reports, we computed a CAARS ADHD Index discrepancy score by subtracting the raw scores from the self- and observer-report and took the absolute value of this difference score as an indication of disagreement between the self and the significant other. The discrepancy index was created as a validity measure to assess response inconsistency [64]. Further, we took the normative T-score of 80 on the self-report as a cut score indicating possible symptom overreporting as suggested by the test manual [63]. Internal consistency (Cronbach’s alpha) of the CAARS was excellent and ranged from 0.74 to 0.95 [64].

The Questionnaire on Mental Ability (FLEI; [51,65]) was applied to assess experienced cognitive abilities in daily life situations. The questionnaire included 30 items, each rated on a 5-point Likert scale, addressing cognitive complaints in the domains of attention, memory, and executive functions. A sum score was calculated indicating the severity of cognitive complaints. The internal consistency (Cronbach’s alpha) of the FLEI was excellent (0.94) [65].

The German version of the Beck Depression Inventory-II (BDI-II; [66,67]) was administered to assess the presence and severity of depressive symptoms over the past two weeks. The BDI-II is a self-reported inventory consisting of 21 items, each scored on a 4-point Likert scale. A larger total score on this scale indicates a higher severity of depressive symptoms. In addition to its interpretation of depressive symptom severity, we considered any score equal to or larger than 38 as possible symptom overreporting [19,68]. The internal consistency (Cronbach’s alpha) of the BDI-II was reported to be high (alpha ≥ 0.84) [69].

The German short version of the Wender Utah Rating Scale (WURS-K; [70,71]) was administered to quantify self-reported ADHD symptoms retrospectively for childhood. The scale includes 25 items (21 symptom items and four control items) on a 5-point Likert scale ranging from 0 (not at all or very slightly) to 4 (very much). The participants were asked to rate each item based on their recall of childhood experiences. Internal consistency (Cronbach’s alpha) of this scale was excellent and reported to be 0.91. The total score on this scale represents the severity of ADHD symptoms in childhood.

#### 2.2.2. Neuropsychological Assessment with Cognitive Tests

A computerized test battery (i.e., CFADHD; [51]) was administered that assesses a range of cognitive functions in which adults with ADHD have been shown to commonly present difficulties. The composition and psychometric properties of the test battery are extensively described in the test manual [52]. Since its introduction, CFADHD has been regularly the subject of neuropsychological research in adult ADHD (e.g., [59,60,61,72,73]). Therefore, it is a presumably well-suited candidate for embedded validity testing. It would be a time- and resource-efficient manner to combine clinical assessment of functioning with validity assessment using the CFADHD. In recent research [50], an analogue study evaluated its use for validity testing and derived cut scores for embedded performance validity (see Table 2). The present study employed eight of the tests as described below, with a total of 17 EVIs.

Processing speed and flexibility were assessed with the Trail-Making Test—Langensteinbach Version (TMT-L; [74]). In Part A, the numbers 1 to 25 were presented simultaneously in a pseudo-random order on the screen, and the participants were asked to tap the numbers in ascending order as quickly as possible. In Part B, the numbers 1 to 13 and the letters A to L were simultaneously and pseudo-randomly presented on the screen. Participants were asked to connect numbers and letters alternately in ascending order as quickly as possible. The time (in seconds) required for part A was used as a measure of processing speed, while the time on part B was interpreted as a measure of mental flexibility.

Selective attention was assessed with the Perceptual and Attention Functions: Selective Attention (WAFS; [75]). In this test, participants were presented with a total of 144 geometric stimuli (triangles, circles, and squares), which could become either darker or lighter, or stay the same. Participants were asked to press the button in response to 30 relevant stimuli (i.e., a circle becomes darker, a circle becomes lighter, a square becomes darker, and a square becomes lighter) as quickly as possible while ignoring irrelevant stimuli. Stimulus presentation time was 1500 milliseconds, with a possible change occurring after 500 milliseconds. The interstimulus interval was 1000 milliseconds. Outcome measures included the mean reaction time (RT) in milliseconds, the standard dispersion of reaction time (SDRT; i.e., the logarithmic standard deviation of the RT), the number of commission errors (CE; i.e., the number of reactions to false or non-existent stimuli), as well as the number of omission errors (OE; number of stimuli with no response). The internal consistency (Cronbach’s alpha) of the main variables was reported to be 0.95.

Working memory was assessed with the two-back paradigm of the N-Back Verbal (NBV; [76]). The participants were presented with a sequence of 100 consonants with a presentation time of 1500 milliseconds each, followed by an inter-stimulus interval of 1500 milliseconds. The task was to press the response button if the consonant currently displayed was identical to the consonant that appeared two places back. The number of correct responses was registered (N). The internal consistency (Cronbach’s alpha) of the main variable correct responses was reported to be 0.85.

Figural fluency was assessed with the 5-point Test—Langensteinbach Version [77]. Squares are presented to the participant, each containing five dots (like the number five on a dice). The participants were requested to connect two or more dots with straight lines and create as many unique patterns as possible within 2 mins. The number of unique patterns created in 2 mins was registered (R). The internal consistency (Cronbach’s alpha) of this variable was reported to be 0.86.

Task switching was assessed with the SWITCH [78]. The task requirement of attention switching was operationalized by bivalent stimuli that could be classified according to two dimensions, i.e., by shape (triangle/circle) or brightness (gray/black). Participants were asked to make categorization judgments based on shape or brightness, with the dimension focus being changed every two stimuli. The items that required a response in the same dimension as the immediately preceding item were defined as repeated items, while items that required a response in a different dimension compared to the preceding item were defined as switching items. The time interval between two items was 750 milliseconds. Outcome measures included task switching accuracy (A; i.e., the difference between the percentage of correct responses for switching and repeated items) and task switching speed (S; i.e., the difference between the mean reaction times for switching and repeated items). The internal consistency of this variable was reported to be high (alpha ≥ 0.81).

Vigilance was assessed with the Perceptual and Attention Functions: Vigilance (WAFV; [79]. In this test, participants were presented with a total of 900 squares that sometimes darkened. Participants are required to respond to 50 target stimuli (the squares becoming darker) by pressing the response button as quickly as possible and ignoring other distracting stimuli. The stimulus presentation time was 1500 milliseconds; a change may occur after 500 milliseconds, and the interstimulus interval was 1000 milliseconds. Outcome measures included the RT in milliseconds, the standard dispersion of reaction time (SDRT), the number of commission errors (CE), and the number of omission errors (OE). The internal consistency (Cronbach’s alpha) of the main variables was reported to be 0.96.

Response inhibition was assessed with the Go/No-Go test [80]. In this test, participants were presented with a series of circles (48 No-Go trials) and triangles (202 Go trials), which were displayed for 200 milliseconds with an interstimulus interval of 1000 milliseconds. Participants were asked to press the response button on triangles and ignore circles. The number of commission errors (CE) was recorded. The internal consistency (Cronbach’s alpha) of this test was reported to be 0.83.

Interference was assessed with the Stroop Interference Test [81]. Key conditions were the reading-interference condition (i.e., requiring participants to react to the meaning of the color-words, e.g., BLUE, GREEN, YELLOW, RED, while ignoring the color of the word, e.g., GREEN printed in red ink), and *naming-interference condition* (i.e., recognition of the color of the word while ignoring the meaning of the word). Outcome measures included *reading interference* (RI) and *naming interference* (NI) by measures of response times in seconds. The internal consistency (Cronbach’s alpha) of the main variables was reported to be 0.97.

### 2.3. Statistical Analysis

Neuropsychological test performances on eight tests, 17 test variables, and their corresponding 17 EVIs were presented and analyzed in descriptive statistics, including measures of central tendency and variation, summary scores, and (cumulative) frequencies. EVI cut scores were applied as presented in Table 2 and as derived and presented in earlier research [19,50,63,68]. Further, EVI failure was analyzed on the test level if at least one EVI showed a positive result, and on the level of the battery if at least one EVI of the entire battery indicated a positive result. In order to explore the association between suspect performance on a particular EVI and suspect performance on the remaining battery, we calculated odds ratios (OR) and their confidence intervals (CIs) for each EVI. Per EVI failure, ORs at, larger, or smaller than 1 were interpreted as no association with any EVI failure in the remaining battery, increased likelihood of another EVI failure, or decreased likelihood of another EVI failure, respectively [82]. The CI gives information on the certainty of the existence of the true effect, e.g., if the null effect OR = 1 can be excluded with sufficient certainty. The association between symptom overreporting (symptom validity testing) and cognitive underperformance (performance validity testing) was explored by the Chi-square test. Cognitive underperformance is defined based on empirical findings and current practice standards by ≥2 positive EVI results in the neuropsychological battery, and, in a corresponding fashion, symptom overreporting is defined by positive results of the validity indices of the CAARS-Self-Report ADHD Index as well as the BDI-II. Effect sizes were indicated by Cramer’s *V* coefficient, which ranged from 0, indicating no association, to 1, indicating perfect association [83]. Finally, we computed a negative binomial regression model in order to determine the explanatory value of clinical measures including symptom self- and other reports (i.e., CAARS-SR-index, CAARS-OR-index, Discrepancy-CAARS-Index, BDI-II, FLEI, WURS-K, ADHD diagnostic status) for the number of suspect results on the 17 EVIs of this battery. Negative binomial regression was chosen because the Poisson-Gamma mixture distribution [84] on which negative binomial regression is based is well suited for predicting count-based data, i.e., non-negative integer values [85,86]. This regression model reports on the regression equation, the goodness of fit, confidence limits, likelihood, and deviance. It performs a comprehensive residual analysis and a subset selection search to find the best regression model with the fewest independent variables [83]. This model was employed because the dependent variable was over-dispersed, which means that the variance (σ^2^) of the dependent variable was greater than the mean (M; M = 1.37, σ^2^ = 3.184). Further, the variance inflation factor (VIF < 10) indicated no evidence for substantial multicollinearity among the independent variables. The majority of the analysis was calculated with IBM SPSS (Version 25.0 for Windows), with the exception of the negative binomial regression, for which we employed R Studio version 4.3.1 [87].

## 3. Results

### 3.1. Base Rates of Positive EVI Results across the Battery

Derived from eight neuropsychological tests, we present base rates of positive EVI results on 17 variables of test performance. Table 3 shows descriptive results of neuropsychological test performance as well as base rates of positive results per EVI and test. Test performance was considered suspect if at least one EVI of the test showed a positive result. Base rates of suspect performance per test ranged from 4.8% (response inhibition) to 23.0–25.4% (selective attention and vigilance) for the total sample (see Figure 1 and Table 3). Considering all 17 EVIs individually, the lowest base rate was shown with 3.3% on the RT of the vigilance test (WAFV-RT), and the highest base rate constituted 17.5% on the same test on the commission errors (WAFV-CE). The base rate of any positive EVI result in the entire battery was 59.1%, indicating that almost 60% of the participants showed at least one suspect result on any EVI in at least one of the eight tests. Figure 1 further shows the EVI failure rates per test in the order of administration, with no remarkable effect of administration time or position in the battery.

Figure 2 and Figure 3 depict cumulative percentages of positive EVI results per test (Figure 2) and test variable (Figure 3). The distributions indicate that more than half of the participants showed at least one positive EVI result across a test variable or test (in both cases 59%), and about 30% of the sample showed suspect results in at least two EVIs across a variable or test. A minority of the participants (<10%) showed suspect results on four or more EVIs per test or five or more EVIs across test variables.

### 3.2. Association between Validity Indicators

Further, the association between the failure on a particular EVI and the failure on at least one more of the remaining EVIs in the battery was indicated by odds ratios (OR). Results, as presented in Table 3, indicate that for the majority of variables, a positive result on one of the EVIs significantly increased the likelihood that this individual showed at least one more positive EVI result in the remaining battery (OR > 1), with CIs not including the null effect for most of the variables. The strongest effect was observed for the vigilance test (RT), with OR = 14.58, 95% CI = 1.901–111.855. An effect in the opposite direction (i.e., a lower likelihood of failing at least one more EVI) was only observed for the SWITCH-S; however, a null effect could not be excluded with sufficient certainty (OR = 0.74, 95% CI = 0.340–1.624).

An analysis of the association between cognitive underperformance and symptom overreporting indicated that 2.8% of individuals failed at both PVT and SVT assessment, while the vast majority (66.8%) showed concordantly valid cognitive performance and symptom reports across both forms of validity assessment. Among those with noncredible performance and/or symptom reports, a larger number of participants showed indications for cognitive underperformance (28.2% of the entire sample) than symptom overreporting (7.8% of the entire sample). In the 30.4% of the overall sample in which cognitive underperformance and symptom overreporting were discrepant, participants were most likely to show cognitive underperformance with no symptom overreporting (25.4%), rather than symptom overreporting with no cognitive underperformance (5.0%). The association between measures of symptom and performance validity is depicted in Table 4. In this Chi-square analysis, symptom overreporting (suspect results on two self-report EVIs) was non-significantly associated with cognitive underperformance (positive EVI results ≥ 2), *χ^2^* = 1.196, *df* = 1, *p* = 0.274, with an effect size (Cramer’s *V* = 0.051) in the small range.

### 3.3. Prediction of Noncredible Test Performance

A negative binomial regression model was computed in order to determine the predictive value of a range of clinical variables for the number of positive results in the 17 EVIs. Table 5 shows the estimates of model parameters, standard errors, Z value, and 95% confidence interval (CI) by profiling the likelihood function and goodness-of-fit statistics such as logarithmic likelihood and Akaike information criteria (AIC). The model shows only one significant predictor for the number of positive EVI results (*p* < 0.05), with the severity of current ADHD symptoms in the self-report being associated with fewer EVIs failures. At the significance level of 5% (−0.0306; 95% CI: −0.0610, −0.0009), the measurement scores of CAARS-SR were negatively related to the number of positive EVI results. Other variables did not exert a significant effect, including other measures of symptom self-report, other-report, a discrepancy of self- and other-report, and ADHD diagnostic status.

## 4. Discussion

The primary aim of this study was to establish base rates of noncredible performance per neuropsychological function and test (variable) on a mixed neuropsychiatric sample by using 17 embedded PVTs. The outcomes of this study will facilitate the assessment of noncredible performance in the clinical assessment of adult ADHD and an understanding of their characteristics and relationships.

### 4.1. EVI Failure Rates per Test and Test Variable

Tests for attention and concentration indicated the highest base rates of noncredible performance. Performance validity test failure rates in the present neuropsychiatric sample of individuals evaluated for adult ADHD ranged from 4.8% (response inhibition) to 23.0–25.4% (selective attention and vigilance; see Figure 1). Failure rates per test variable ranged from 3.3% on the reaction time of the vigilance test (WAFV-RT) to 17.5% on the commission errors of the vigilance test (WAFV-CE), which falls within the range, though at the lower end, of the estimated base rates of single PVT failures in clinical assessments of adults with ADHD reported in previous studies [22,23,24]. Estimated base rates in earlier studies varied within a broad range and are difficult to compare with each other because they seem to depend on various factors, including the specific PVT measure applied, embedded or stand-alone assessment (with higher sensitivity of stand-alone measures, see for example [24,88,89]), referral context (e.g., ADHD diagnosed or mixed neuropsychiatric samples, see for example [1]), or sample characteristics (with higher base rates in student samples, see for example [90,91]). On a cautionary note, higher rates of noncredible cognitive performance in this study compared to some of the previous research may be explained by the larger number of EVIs. The consideration of a large number of EVIs bears the risk of inflating false positive findings and may require adjustments of the number of EVI failures that are defined to indicate noncredible cognitive performance (for a recent discussion, see [92]). Also conforming to earlier findings, tests of selective attention and vigilance were most useful in this context based on the observation of higher failure rates [42,44,50,93]. However, because these tests proved to be most sensitive in their development as EVIs (sensitivities of 63–65%, compared to 0–56% on the remaining tests of the battery, see [50]), it cannot be concluded whether the present data indeed give support for tests for selective attention and vigilance as the most sensitive in detecting noncredible performance, or whether a higher proportion of individuals underperformance on those tests compared to the remaining tests of the battery. In support of the latter explanation, it can be argued that careless examinees may most likely underperform on (long-lasting) tests for attention because of their monotonous character, and individuals deliberately feigning ADHD may decide in particular on attention tests to perform below their ability levels because these tests may appear as if they may be relevant to assess core symptoms of ADHD. Nowadays, ADHD and its behavioral characteristics are regularly presented in various forms of media to the general public. The dominance of attention problems is, therefore, known to most people, which may motivate individuals deliberately feigning ADHD to underperform, particularly in attention tests. Further, attention tests allow a fine-grained behavioral analysis, including the assessment of task accuracy, i.e., errors of omission and commission, task speed, variability of speeded reactions, and its trade-off. Based on this nuanced assessment, it can be assumed that attention tests are well suited to distinguish those putting forth the best effort to perform well and those showing noncredible performance. Furthermore, almost 60% of our participants showed at least one suspect result on any EVI in any of the eight tests. Based on the stricter and currently widely accepted criterion of defining noncredible performance by at least two PVT failures, we yielded a base rate of noncredible performance of about 30% per test variable (32.6%) or test (28.3%). This number is higher and outside the range compared to previous studies using the same criterion of determining invalid cognitive performance (≥2 PVT failures; e.g., 9–19%; [13,26,27]). The reason for this difference may be found in the number of applied PVTs, as base rates of positive results on two or more PVTs increase with the number of measures applied in the respective battery, and may be an indication of normal performance variability rather than invalid performance [94]. While the present study counted the number of EVI failures in a large battery of eight tests, comprising 17 EVIs, the majority of previous studies derived their base rate estimations from batteries with one to seven embedded PVTs [26,48,53,90,95]. To account for the higher number of measures in the present study and avoid confusion of normal performance variability with invalid performance, a stricter criterion for invalid performance may be applicable, e.g., positive results on four or more EVIs, which would result in a 10% base rate of noncredible performance and corresponding base rates estimated in previous research. Across the different criteria applied, we show non-trivial and substantial base rates of noncredible cognitive performance that emphasize the importance of validity assessment in real-world clinical settings in order to facilitate and support accurate clinical diagnoses, treatment planning, and evaluation [96].

### 4.2. The Effect of Test Administration Order on EVI Failure Rates

Inspecting EVI failure rates across the order of test administration does not show any seemingly relevant effect of time on task. In other words, there is no indication of a systematic fluctuation of test compliance and effort across this long battery of eight neuropsychological tests. However, this finding must be interpreted cautiously since the test administration order was fixed and not randomized. Because tests differed in their characteristics, and accuracy in differentiating credible from noncredible performance, the inspection of the effects of administration time on performance validity does not allow a firm conclusion. The examination of EVI failure across a long battery needs to be examined in systematically planned and controlled studies in order to empirically support the claim of sampling performance validity continuously throughout an assessment and across cognitive domains.

### 4.3. Associations within EVIs and between SVTs and PVTs

Further, occurrences of EVI failures seem to be highly associated with each other, as shown by ORs larger than one for the vast majority of variables, with up to a 15-times higher likelihood of failing any other EVI when showing a positive result on the reaction time of the vigilance test (WAFV-RT).

In contrast, SVT and PVT assessment showed only limited correspondence and seemed to represent largely different forms of validity assessment in this heterogeneous neuropsychiatric sample, as shown by a nonsignificant association of small size and discrepant classification. In contrast to the substantial number of individuals showing cognitive underperformance, only a small proportion of individuals showed symptom overreporting, which resulted in the discrepant classification of about 25% of individuals showing indication for cognitive underperformance with no symptom overreporting. These results suggest that cognitive underperformance and symptom overreporting tests measure distinct but related constructs. Therefore, both types of validity tests are needed in clinical ADHD neuropsychological assessment, in order to support the validity of both self-report questionnaires and performance tests. The strength of the association between measures of symptom and performance validity in previous research appears to vary depending on population, assessment context, and applied measures. In neuropsychiatric patients undergoing clinical evaluation of adult ADHD, PVTs and SVTs have been shown to be rather dissociable [30,97,98], whereas more concordance has been shown in disability claimants [99]. Giromini et al. [100] brought forward several explanations that may explain this inconsistency in the findings on the SVTs/PVTs relationship, including the relatively few validated SVTs compared to PVTs, differences in optimal SVTs cut scores across populations, differences in agreed standards for determining invalidity (i.e., ≥2 independent PVT failures but no similar standards agreed on SVTs application, yet), and PVTs being generally evaluated as a more unitary underlying construct compared to SVTs. In future research, the relationship between symptom validity and performance validity in individuals with ADHD and other clinical groups needs to continue to be clarified, as this is a common issue in neuropsychological assessment in general regarding the discrepancy between performance-based and self-report measures. The discrepancy may be explained by differences in conceptual nature reflecting optimal performance under clear instructions within a short period of time (“what I can do” as assessed by performance tests) vs. typical functioning in real-life conditions with no clear rules and instructions but where one’s own priorities and goals need to be set (“what I do” as assessed by self-report forms) [101,102,103].

### 4.4. Prediction of Cognitive Underperformance

We demonstrate that clinical characteristics have no meaningful predictive value for cognitive underperformance. Cognitive underperformance occurs in a substantial number of cases; however, the reasons for invalid test data are poorly understood and were explained by a heterogeneous set of overlapping factors [104]. Prediction models of the present study, based on negative binomial regression, found only weak evidence for characterizing cognitive underperformance. Among the set of predictors, only the severity of current ADHD symptoms fell just below the threshold, indicating significance for predicting the number of EVI failures. However, despite reaching significance, the effect may not be of clinical relevance, as the upper bound of the CI falls close to zero, which may indicate that a trivial effect cannot be excluded with sufficient certainty. PVT failure does not seem to be explained sufficiently well by clinical characteristics, which is in line with earlier findings on large samples of a comparable referral context (see for example, [3,28]). In the present study, individuals referred to the outpatient diagnostic unit had to accept several months of waiting time before they were invited for an assessment, which suggests that "help-seeking behavior" may be a reason for noncredible data in at least some of the cases in order to convince the clinician of their experienced problems and need for support. This behavior is only one possible explanation for noncredible data, and many other, internally or contextually defined, factors may contribute to poor symptom and performance validity. It must be noted, however, that this explanation was critically discussed in recent research because it is difficult to operationalize and may be impossible to falsify (see [4], for a graphical overview of different explanatory levels). More work is needed to study underlying motivation and distinguish between different reasons for underperformance, including malingering, careless behavior, excuse-making behavior, or unconscious (i.e., unintentional) forms of underperformance. Repeated assessments and large-scale longitudinal studies may be appropriate means to shed light on this question by following individuals with noncredible symptom reports and/or test performance and comparing their clinical trajectory and outcomes to those with credible symptom reports and test data. Furthermore, another potential reason for the weak association between clinical characteristics (mainly assessed by self- and other-report rating scales, and diagnostic status) and cognitive underperformance (number of noncredible results across the entire battery) is the discrepancy between performance-based and self-report measures (as explained above for the SVTs and PVTs relationships). Self- and other- reports, and diagnostic status, may reflect typical functioning in real-life conditions where no clear rules and instructions are given, but where the patient’s own priorities and goals need to be set (“what I do”). In contrast, EVI test failure in a large neuropsychological battery reflects test performance under clear instructions (“maximize performance”) within a short period of time. Weak associations between cognitive test performance and self- or informant ratings have been reliably demonstrated in previous research on children and adults with ADHD [101,102,103].

### 4.5. Strength, Limitations, and Future Directions

This study has several methodological strengths, including a large, naturalistic clinical sample, a large number of EVIs, and the examination of performance validity across cognitive domains and along a two-hour neuropsychological assessment. However, several study limitations must be considered. First, the present study does not include established stand-alone PVTs to contrast embedded indicators of performance validity. The inclusion of stand-alone validity measures would have been useful, i.e., as criterion measures, as their sensitivities are usually higher, and they are traditionally considered the benchmark instruments in performance validity assessment. We stress that a combination of the use of multiple embedded and stand-alone PVTs is most appropriate for identifying underperformance with the greatest confidence in future research. Second, the present study employs EVIs that have been developed on and derived from an analogue study and are still pending validation against the independent classification of credibility in clinical practice. Validation in a known-groups comparison is relevant to advise clinical application because simulation designs are generally criticized for their limited generalizability to actual, real-world malingerers [104,105,106]. Further, the original analogue study determined EVI cut scores based on the (simulated) performance of university students, which differs greatly from the characteristics of the present clinical sample. Although validity measures are known to be relatively insensitive to age and education variables, embedded measures may be more sensitive because they are derived from original performance measures. Further, more research is needed to replicate our findings in different clinical or non-clinical referral contexts that make use of standardized neuropsychological assessments. Research using known-group comparisons appears particularly helpful in evaluating the appropriateness of the cut scores used in the present and previous studies. Third, the test presentation order of the administered battery was not randomized but fixed for all participants, which may exert an effect on test performance and EVI results (e.g., due to novelty, fatigue, and test motivation), and may thus hamper a valid comparison of EVIs. Finally, and fourth, it must be stressed that the effects of time on tasks (e.g., EVI failure) are difficult to interpret because not all tests are equally sensitive. Thus, differences in EVI failure rates across time could be caused by either time or alternative factors such as test sensitivity or task characteristics.

## 5. Conclusions

The present study on a naturalistic sample of individuals undergoing clinical evaluation of adult ADHD provides base rate estimations of about 10 to up to 30% noncredible test performance on a large, two-hour battery of neuropsychological testing. PVT failures occurred in a sizeable number, across the entire assessment, and seemed to represent a coherent construct. Tests for attention appeared most adequate and sensitive, requiring further clinical validation. We conclude, and support the findings of earlier research, that performance validity assessment is imperative for adequate clinical assessment, is nonredundant from symptom validity assessment, and cannot be predicted by most standard clinical routine measures. These results further emphasize the importance of administering multiple (embedded) PVTs during clinical assessments and supporting the clinician in the application and interpretation of test data of this or related batteries. It remains a subject for research to determine the optimal number of positive EVIs that reliably indicate invalid cognitive performance in large batteries containing many EVIs, in order to prevent inflation of false positives. Future studies should further examine how these results would relate to application in other clinical and non-clinical populations and differentiate between underlying motivations.

## Figures and Tables

**Figure 1 jcm-12-06926-f001:**
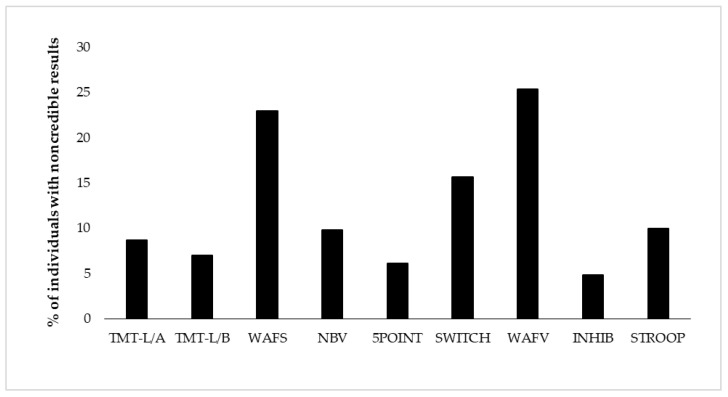
Proportion of individuals showing positive EVI (embedded validity indicator) results per test (i.e., positive EVI (embedded validity indicator) result on at least one of its variables). The order of display represents the test order in the battery. TMT-L/A = Trail-Making Test-L, Part A, TMT-L/B = Trail-Making Test-L, Part B, WAFS = Perceptual and Attention Functions Test-Selective, NBV = N-back verbal, 5POINT = 5-Point, SWITCH = Task switching, WAFV = Perceptual and Attention Functions-Vigilance, INHIB = Go/No-Go test, STPOOP = Stroop interference test.

**Figure 2 jcm-12-06926-f002:**
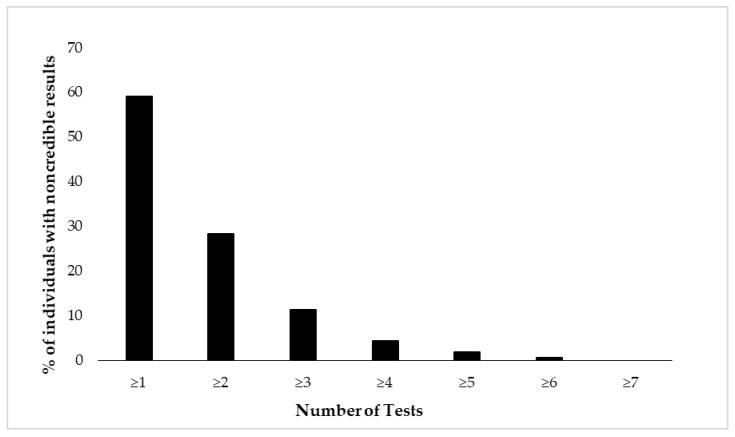
Cumulative percentage of individuals (N = 462) showing suspect results in the neuropsychological battery of eight tests. Test performance is considered suspect if at least one of its variables indicates positive EVI result.

**Figure 3 jcm-12-06926-f003:**
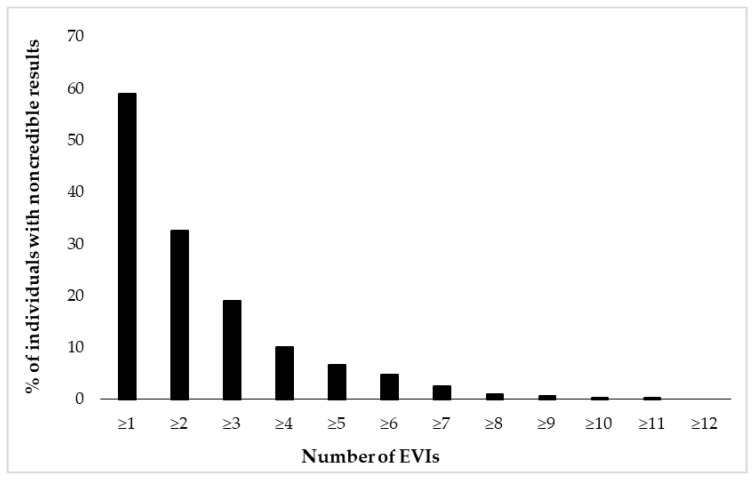
Cumulative percentage of individuals (N = 462) showing suspect results in 17 embedded validity indicators (EVIs).

**Table 1 jcm-12-06926-t001:** Demographic characteristics of participants (N = 464).

Adults with Suspected ADHD (N = 464)					
Gender (Male/Female)	275/189				
Education level (% in 1/2/3/4/5/6) ^a^	2.2/19.5/31.2/30.1/17.1/2.2				
Stimulant medication (Yes/No)	9/455				
	Min	Max	Median	Mean	SD
Age (in years)	18	62	32	34.1	10.8
CAARS-SR- index ^b^	3	36	23	22.2	6.4
CAARS-OR-index ^b^	0	56	20	19.8	7.2
Discrepancy-CAARS-Index ^c^	0	25	5	6.1	5.4
BDI ^d^	0	54	20	21.3	11.8
FLEI ^e^	2	120	78	75.2	20.1
WURS-K ^f^	3	75	35	35.7	14.1

^a^ Education (1/2/3/4/5/6) = no school-leaving qualification/compulsory schooling or secondary school completed/completed technical school or vocational training/higher school with university entrance qualification/university or college degree/not reported. ^b^ Self-Reports (SR) or Other Reports (OR) of Conners’ Adult ADHD Rating Scales; Index, ADHD index. ^c^ Absolute discrepancy between self- and other reports of ADHD symptoms. ^d^ Beck Depression Inventory-II. ^e^ Questionnaire on Mental Ability. ^f^ Wender Utah Rating Scale for childhood ADHD symptoms.

**Table 2 jcm-12-06926-t002:** Embedded validity indicators (EVIs) and their cut scores in the neuropsychological battery.

Function	Test/Questionnaire	EVI	Cut-off Score ^a^
Performance Validity			
Processing speed	Trail-Making Test-L, Part A	Response time (RT)	≥27.1
Cognitive flexibility	Trail-Making Test-L, Part B	Response time (RT)	≥54.6
Selective attention	Perceptual and Attention Functions Test-Selective (WAFS)	Response time (RT)	≥532
		Dispersion of response times (RTSD)	≥1.43
		Omission errors (OE)	≥4
		Commission errors (CE)	≥8
Working memory	N-back verbal (NBV)	Number of correct responses (N)	≤6
Figural fluency	5-Point (5POINT)	Number of correct responses (R)	≤12
Task switching	SWITCH	Accuracy (A)	≥11
		Speed (S)	≥0.54
Vigilance	Perceptual and Attention Functions-Vigilance (WAFV)	Response time (RT)	≥636
		Dispersion of response times (RTSD)	≥1.35
		Omission errors (OE)	≥7
		Commission errors (CE)	≥7
Response inhibition	INHIB (Go/No-Go)	Commission errors (CE)	≥27
Interference control	STROOP	Reading interference (RI)	≥0.448
		Naming interference (NI)	≥0.323
Symptom Validity			
ADHD symptoms	Conners’ Adult ADHD Rating Scales	CAARS-Self-Report-Index	>80 ^b^
Depressive symptoms	Beck Depression Inventory-II	Total score	≥38 ^c^

^a^ Raw score. Cut scores determined by Becke et al. [50]. ^b^ T-score. Cut score suggested in the test manual by Conners et al. [64]. ^c^ Cut score based on Fuermaier et al. [19].

**Table 3 jcm-12-06926-t003:** Neuropsychological test performance and results for their embedded validity indicators (EVIs).

Test/Variable	N	Range (Min–Max)	Median	Mean	SD	% Positive EVI	Odds Ratio	95% CI
TMT-L		-	-	-	-	11.7	3.05	(1.559, 5.960)
A-RT	461	2.2–191.0	19.2	20.5	9.9	8.7	2.49	(1.188, 5.227)
B-RT	460	3.8–110.3	29.3	32.6	13.4	7.0	2.46	(1.082, 5.608)
WAFS		-	-	-	-	23.0	3.89	(2.359, 6.416)
-RT	461	0–739	349	362.3	86.3	3.7	6.79	(1.535, 30.049)
-RTSD	461	0–9.8	1.3	1.3	0.7	7.8	2.06	(0.988, 4.290)
-OE	461	0–30	0	0.9	2.5	6.7	3.15	(1.330, 7.473)
-CE	461	0–81	3	4.6	7.3	15.6	6.77	(3.371, 13.584)
NBV		-	-	-	-	9.8	2.06	(1.050, 4.035)
-N	460	0–23	12	11.3	3.4	9.8	2.06	(1.050, 4.035)
5POINT		-	-	-	-	6.1	3.58	(1.335, 9.587)
-N	460	6–59	25.5	26.4	9.8	6.1	3.58	(1.335, 9.587)
SWITCH		-	-	-	-	15.7	1.07	(0.645, 1.766)
-A	459	−19–32	3	3.6	5.6	9.8	1.33	(0.714, 2.476)
-S	459	−0.886–0.946	0.2	0.2	0.2	5.9	0.74	(0.340, 1.624)
WAFV		-	-	-	-	25.4	2.67	(1.712, 4.171)
-RT	456	28–726	447	454.0	83.4	3.3	14.58	(1.901, 111.855)
-RTSD	451	0–8	1	1.1	0.5	4.7	1.64	(0.666, 4.037)
-OE	456	0–24	2	3.0	3.9	14.5	3.33	(1.851, 5.983)
-CE	456	0–154	3	5.1	13.6	17.5	3.64	(2.107, 6.271)
INHIB		-	-	-	-	4.8	1.11	(0.466, 2.662)
-CE	456	0–34	14	14.5	7.2	4.8	1.11	(0.466, 2.662)
STROOP		-	-	-	-	10.0	4.03	(1.833, 8.841)
-RI	457	−0.062–1.651	0.2	0.2	0.2	5.9	6.58	(1.951, 22.168)
-NI	457	−0.070–0.931	0.1	0.1	0.1	6.6	3.24	(1.297, 8.080)
Entire battery		-	-	-	-	59.1	-	

CI = Confidence interval, RT = Response time, RTSD = Dispersion of response times, OE = Omission errors, CE = Commission errors, N = Number of correct responses, R = Number of repetitions, A = Accuracy, S = Speed, RI = Reading interference, NI = Naming interference, TMT-L = Trail-Making Test-L, WAFS = Perceptual and Attention Functions Test-Selective, NBV = N-back verbal, 5POINT = 5-Point, SWITCH = Task switching, WAFV = Perceptual and Attention Functions Test-Vigilance, INHIB = Go/No-Go test, STROOP = Stroop interference test.

**Table 4 jcm-12-06926-t004:** Association between symptom overreporting (2 SVTs) and cognitive underperformance (≥2 EVIs).

Cognitive Underperformance	Symptom Overreporting			*χ^2^*	*p* Value	Cramer’s *V*
	Yes	No	Total			
Yes	13 (2.8%)	118 (25.4%)	131 (28.2%)	1.196	0.274	0.051
No	23 (5.0%)	310 (66.8%)	333 (71.8%)			
Total	36 (7.8%)	428 (92.2%)	464 (100%)			

Cognitive underperformance is defined by positive results on ≥2 EVIs. Symptom overreporting is defined by suspect results on both the CAARS-Self-Report ADHD Index and BDI-II.

**Table 5 jcm-12-06926-t005:** Negative binomial regression models based on clinical measures to predict the number of positive EVI results (N = 337).

Variables	Estimate	Std. Error	Z Value	95% CI	*p* Value
CAARS-SR- index ^a^	−0.0306	0.0156	−1.967	(−0.0610, −0.0009)	0.0492 *
CAARS-OR-index ^a^	0.0247	0.0133	1.856	(−0.0013, 0.0510)	0.0634
Discrepancy-CAARS-Index ^b^	0.0107	0.0163	0.659	(−0.0220, 0.0440)	0.5097
BDI ^c^	0.0089	0.0070	1.272	(−0.0048, 0.0225)	0.2033
FLEI ^d^	0.0059	0.0050	1.183	(−0.0033, 0.0151)	0.2366
WURS-K ^e^	0.011	0.0061	1.796	(−0.0008, 0.0227)	0.0725
ADHD/*n*-ADHD group	0.1021	0.1558	0.655	(−0.2055, 0.4091)	0.5124
Intercept	−0.7616	0.3944	−1.931	(−1.5211, −0.0121)	0.0535
Theta	1.086				
log-likelihood	−1025.698				
AIC	1043.7				

CI = Confidence Interval. ^a^ Self-Reports (SR) or Others Reports (OR) of Conners’ Adult ADHD Rating Scales; Index, ADHD index. ^b^ Absolute discrepancy between self- and others reports of ADHD symptoms. ^c^ Beck Depression Inventory-II. ^d^ Questionnaire on Mental Ability. ^e^ Wender Utah Rating Scale for childhood ADHD symptoms. * Statistically significant at *p* < 0.05. AIC = Akaike information criteria.

## Data Availability

The data that support the findings of this study are available from the corresponding author upon reasonable request.

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
