# Peer review of "Cognitive Underperformance in a Mixed Neuropsychiatric Sample at Diagnostic Evaluation of Adult ADHD"

_jcm, 2023, doi:10.3390/jcm12216926_

Round 1
Reviewer 1 Report
Comments and Suggestions for Authors
Abstract
- The abstract is somewhat lengthy, making it challenging for readers to grasp the main points quickly. It should be condensed to provide a concise summary of the study.
- The abstract could benefit from better organization. It should follow a clear structure, including Background, Objectives, Methods, and Results/Conclusions, each with a brief, focused description.
- While the objectives are briefly mentioned, they could be more specific and clearly stated. Instead of saying "examines indicators of cognitive underperformance," specify what aspects of cognitive underperformance the study investigates.
- The description of the methods could be more detailed. Mention the specific neuropsychological tests used, as this is crucial for replicability and transparency in research.
- The abstract's conclusion is somewhat vague. Provide more specific insights into the implications of the findings and potential avenues for future research.
Introduction
- The introduction is quite lengthy and contains a significant amount of information. Consider breaking it into smaller sections or paragraphs to improve readability. Clearly label each section to guide readers through the content.
- While comprehensive, the introduction can be dense and complex at times. Aim for greater clarity and conciseness. Avoid overly long sentences and complex phrasing.
- Explicitly state the research objectives and any hypotheses you intend to test in your study. This will provide readers with a clear understanding of what you aim to achieve.
- Please use the below references for enrichment of your introduction:
1. https://journals.plos.org/plosone/article?id=10.1371/journal.pone.0277175
2. http://eprints.lums.ac.ir/3290/
3. https://www.sciencedirect.com/science/article/abs/pii/S0278584622001051
- While you mention the significance of the study briefly, consider expanding on this aspect. Explain in more detail why this research is important and how it contributes to the existing body of knowledge.
- Define any specialized terms, abbreviations, or acronyms upon first use to ensure that readers, including those new to the field, can follow your argument.
- Use transition sentences to smoothly guide readers from one topic or section to the next. This will improve the flow of your introduction.
- Consider using thematic subheadings within the introduction to signal shifts in the focus or topic. For example, you could have subheadings like "Noncredible Performance in ADHD Assessments," "Importance of Validity Measures," "Previous Research Findings," and so on.
- Begin your introduction with a compelling hook or anecdote that captures the reader's interest. This can help make your research more engaging and accessible.
Materials and Methods Section
· The description of participant recruitment is clear, including the source of recruitment and diagnostic criteria. However, it would be helpful to specify the time frame during which participants were recruited to understand if there were any changes in diagnostic criteria or clinical practices over time.
· Mention if there were any inclusion/exclusion criteria beyond suspected ADHD that were used to select participants. This information would help readers understand the participant sample better.
· While you mention that participants provided written informed consent, it would be beneficial to include more details about the informed consent process, such as whether participants were informed about the research objectives, risks, and benefits.
· Provide additional information about the ethical approval process, including the date of approval and any ethical considerations specific to this study.
· The description of participant demographics and clinical characteristics is informative. However, consider presenting this information in a more concise and reader-friendly format, such as a table, to make it easier to grasp the key statistics.
· The description of the materials, including the questionnaires and neuropsychological tests, is clear. However, it would be helpful to provide references for these materials, especially if they have been modified or adapted for this study.
· If there were any specific versions or adaptations of the tests/questionnaires used, please specify them to ensure transparency.
· The explanation of the statistical analysis is sufficient for readers with a background in statistics. However, it might be beneficial to briefly explain the rationale behind choosing specific statistical methods, particularly negative binomial regression, for those less familiar with statistics.
· Consider providing information about the software or programming languages used for statistical analysis (e.g., R Studio), along with version numbers.
· The section could benefit from subheadings to clearly delineate different aspects of the materials and methods, making it easier for readers to navigate the content.
Discussion
v The discussion section could benefit from more explicit subheadings to delineate the different objectives and findings. Clear subheadings would help guide the reader through the discussion more effectively.
v The introductory paragraph is somewhat lengthy and could be condensed to provide a more concise overview of the study's objectives.
v The discussion adequately addresses each of the study's objectives, providing insights into base rates of noncredible performance, test sensitivity, associations between symptom and performance validity, predictors of noncredible performance, and methodological considerations.
v Consider adding a brief summary at the beginning of each subsection to reinforce the key points related to each objective.
v The discussion effectively compares the study's findings with previous research, which adds depth to the interpretation of results.
v To further strengthen the discussion, provide specific citations to relevant studies or literature when making comparisons or referencing previous findings. This would lend greater credibility to the claims made.
v While the discussion mentions the sensitivity of selective attention and vigilance tests in detecting noncredible performance, it lacks concrete evidence or references to support these claims. Consider citing studies that have established the sensitivity of these tests in similar contexts.
v Provide more detailed explanations for why certain tests might be more sensitive to noncredible performance, supported by relevant research or theoretical frameworks.
v The discussion effectively highlights the limited correspondence between symptom validity tests (SVTs) and PVTs in the study's population, which aligns with previous research.
v Continue to explore implications of this discrepancy. How might this impact clinical assessment practices? Are there potential solutions or alternative methods worth discussing?
v The discussion mentions weak evidence for characterizing cognitive underperformance based on clinical characteristics. Delve deeper into potential reasons why clinical characteristics might not be strong predictors.
v When discussing the hypothesis of "help-seeking behavior" contributing to noncredible data, consider presenting it as speculative idea rather than definitive conclusion. Include suggestions for future research that could explore this hypothesis.
v The discussion effectively highlights the methodological strengths of study and acknowledges its limitations, providing a balanced assessment.
v Emphasize the significance of addressing these limitations in future research and how doing so could strengthen the field's understanding of performance validity assessment.
v The conclusion succinctly summarizes key findings and emphasizes importance of performance validity assessment in clinical settings.
v Consider expanding on the implications of the research findings for clinical practice and the potential benefits of addressing noncredible performance.
Comments on the Quality of English Language
Abstract
- The abstract is somewhat lengthy, making it challenging for readers to grasp the main points quickly. It should be condensed to provide a concise summary of the study.
- The abstract could benefit from better organization. It should follow a clear structure, including Background, Objectives, Methods, and Results/Conclusions, each with a brief, focused description.
- While the objectives are briefly mentioned, they could be more specific and clearly stated. Instead of saying "examines indicators of cognitive underperformance," specify what aspects of cognitive underperformance the study investigates.
- The description of the methods could be more detailed. Mention the specific neuropsychological tests used, as this is crucial for replicability and transparency in research.
- The abstract's conclusion is somewhat vague. Provide more specific insights into the implications of the findings and potential avenues for future research.
Introduction
- The introduction is quite lengthy and contains a significant amount of information. Consider breaking it into smaller sections or paragraphs to improve readability. Clearly label each section to guide readers through the content.
- While comprehensive, the introduction can be dense and complex at times. Aim for greater clarity and conciseness. Avoid overly long sentences and complex phrasing.
- Explicitly state the research objectives and any hypotheses you intend to test in your study. This will provide readers with a clear understanding of what you aim to achieve.
- Please use the below references for enrichment of your introduction:
1. https://journals.plos.org/plosone/article?id=10.1371/journal.pone.0277175
2. http://eprints.lums.ac.ir/3290/
3. https://www.sciencedirect.com/science/article/abs/pii/S0278584622001051
- While you mention the significance of the study briefly, consider expanding on this aspect. Explain in more detail why this research is important and how it contributes to the existing body of knowledge.
- Define any specialized terms, abbreviations, or acronyms upon first use to ensure that readers, including those new to the field, can follow your argument.
- Use transition sentences to smoothly guide readers from one topic or section to the next. This will improve the flow of your introduction.
- Consider using thematic subheadings within the introduction to signal shifts in the focus or topic. For example, you could have subheadings like "Noncredible Performance in ADHD Assessments," "Importance of Validity Measures," "Previous Research Findings," and so on.
- Begin your introduction with a compelling hook or anecdote that captures the reader's interest. This can help make your research more engaging and accessible.
Materials and Methods Section
· The description of participant recruitment is clear, including the source of recruitment and diagnostic criteria. However, it would be helpful to specify the time frame during which participants were recruited to understand if there were any changes in diagnostic criteria or clinical practices over time.
· Mention if there were any inclusion/exclusion criteria beyond suspected ADHD that were used to select participants. This information would help readers understand the participant sample better.
· While you mention that participants provided written informed consent, it would be beneficial to include more details about the informed consent process, such as whether participants were informed about the research objectives, risks, and benefits.
· Provide additional information about the ethical approval process, including the date of approval and any ethical considerations specific to this study.
· The description of participant demographics and clinical characteristics is informative. However, consider presenting this information in a more concise and reader-friendly format, such as a table, to make it easier to grasp the key statistics.
· The description of the materials, including the questionnaires and neuropsychological tests, is clear. However, it would be helpful to provide references for these materials, especially if they have been modified or adapted for this study.
· If there were any specific versions or adaptations of the tests/questionnaires used, please specify them to ensure transparency.
· The explanation of the statistical analysis is sufficient for readers with a background in statistics. However, it might be beneficial to briefly explain the rationale behind choosing specific statistical methods, particularly negative binomial regression, for those less familiar with statistics.
· Consider providing information about the software or programming languages used for statistical analysis (e.g., R Studio), along with version numbers.
· The section could benefit from subheadings to clearly delineate different aspects of the materials and methods, making it easier for readers to navigate the content.
Discussion
v The discussion section could benefit from more explicit subheadings to delineate the different objectives and findings. Clear subheadings would help guide the reader through the discussion more effectively.
v The introductory paragraph is somewhat lengthy and could be condensed to provide a more concise overview of the study's objectives.
v The discussion adequately addresses each of the study's objectives, providing insights into base rates of noncredible performance, test sensitivity, associations between symptom and performance validity, predictors of noncredible performance, and methodological considerations.
v Consider adding a brief summary at the beginning of each subsection to reinforce the key points related to each objective.
v The discussion effectively compares the study's findings with previous research, which adds depth to the interpretation of results.
v To further strengthen the discussion, provide specific citations to relevant studies or literature when making comparisons or referencing previous findings. This would lend greater credibility to the claims made.
v While the discussion mentions the sensitivity of selective attention and vigilance tests in detecting noncredible performance, it lacks concrete evidence or references to support these claims. Consider citing studies that have established the sensitivity of these tests in similar contexts.
v Provide more detailed explanations for why certain tests might be more sensitive to noncredible performance, supported by relevant research or theoretical frameworks.
v The discussion effectively highlights the limited correspondence between symptom validity tests (SVTs) and PVTs in the study's population, which aligns with previous research.
v Continue to explore implications of this discrepancy. How might this impact clinical assessment practices? Are there potential solutions or alternative methods worth discussing?
v The discussion mentions weak evidence for characterizing cognitive underperformance based on clinical characteristics. Delve deeper into potential reasons why clinical characteristics might not be strong predictors.
v When discussing the hypothesis of "help-seeking behavior" contributing to noncredible data, consider presenting it as speculative idea rather than definitive conclusion. Include suggestions for future research that could explore this hypothesis.
v The discussion effectively highlights the methodological strengths of study and acknowledges its limitations, providing a balanced assessment.
v Emphasize the significance of addressing these limitations in future research and how doing so could strengthen the field's understanding of performance validity assessment.
v The conclusion succinctly summarizes key findings and emphasizes importance of performance validity assessment in clinical settings.
v Consider expanding on the implications of the research findings for clinical practice and the potential benefits of addressing noncredible performance.
Reviewer 2 Report
Comments and Suggestions for Authors
This interesting study assesses base rates of noncredible performance in clinical evaluations of adult ADHD on one or more of 17 embedded validity indicators (EVIs) and also examines indicators of cognitive underperformance across the administration of a 2-hour 20 battery, its association with symptom overreporting, and its characterization by clinical information. The main findings is that tests for attention and concentration appear to be the most adequate and sensitive for the detection of underperformance in ADHD.
The design and methodology of this study are fine, data are clearly reported and addressed in the Discussion. Statistical analysis is adequate.
Author Response
We thank the reviewer for the positive evaluation.
Round 2
Reviewer 1 Report
Comments and Suggestions for Authors
The topic is fascinating, but a thorough restructuring of the content is necessary. I kindly request the esteemed author to revise the entire article with a deeper and more coherent perspective, taking into consideration the suggested changes. Afterward, please resubmit it for further review. I believe this feedback will be valuable in improving the quality of your work.
Best regards.
Abstract
- Expand on the research objectives and clarify the specific research questions. For example, what is the significance of examining the order of test administration and its relationship with EVI failure rates? How does the prediction of cognitive underperformance by clinical information contribute to the study's goals?
- Mention the significance or potential implications of these objectives for the field of adult ADHD assessment.
- Provide more details about the neuropsychiatric sample. What are the characteristics of the participants? Mention any inclusion/exclusion criteria.
- Explain why the Vienna Test System (VTS; CFADHD) was chosen for the neuropsychological assessment and why it is suitable for this study.
- Describe the specific neuropsychological tests used for embedded performance validity indicators (PVTs). This will help readers understand the methodology better.
- Mention how the self- and other report symptom rating scales were chosen and why they are appropriate for assessing depressive symptoms and cognitive functioning.
- Elaborate on which specific PVTs or neuropsychological tests were the most sensitive for detecting underperformance. Provide relevant statistical information if available.
- Explain the significance of cognitive underperformance as a coherent construct and how it can influence the diagnostic process.
Introduction
- Provide a more concise and structured overview of the key points. Break down this section into shorter paragraphs for readability.
- Define "noncredible symptom report" and "noncredible performance" more explicitly.
- Discuss the potential impact of noncredible symptom reports and test performance on patients and the healthcare system.
- Cite the specific studies (references [1-3]) to give readers a clearer idea of the evidence in this area.
- Offer examples of specific EVIs derived from cognitive tests and their relevance to ADHD assessment.
- Clarify any variations in sample composition, methodology, or criteria used across the studies that may explain differences in base rates.
- Please for enrichment your introduction about ADHD, use the below references:
http://journalaim.com/Article/aim-20829
https://brieflands.com/articles/ijpbs-108390.html
- Address the exceptions or contradictions in the literature (e.g., White et al. 2022) in more detail.
- Explore the need for further research to compare the value of different embedded validity indicators (EVIs) across multiple cognitive domains.
- Consider discussing the challenges associated with assessing credibility in real-world clinical settings versus controlled studies.
- Clarify the role of the current study in building upon or validating the findings of Becke et al. (2023).
Materials and Methods
- Provide a more detailed description of the demographic characteristics of the participants, including age range, gender distribution, and education levels.
- Suggestion: Explain in more detail how the cognitive tests on the Vienna Test System were chosen and justify why they were considered appropriate for the study.
- Discuss the reliability and validity of the neuropsychological assessment tools and provide relevant references to support their use in this context.
- While conducting retrospective studies is acceptable, provide a more thorough explanation of the ethics protocol and the reasons for seeking ethical permission towards the end of data collection.
- Expand on the characteristics of participants, including their age range, gender distribution, and educational background. This will help readers better understand the diversity of the sample.
- Provide more context for the comorbid psychiatric disorders observed in the participants and their potential impact on the study.
- Include a brief statement about the ethical considerations for recruitment and data collection, explaining how informed consent was obtained.
- Mention the reliability and validity of the questionnaires used to assess ADHD symptoms and other relevant psychopathology.
Results
- Provide more context on what EVIs are, why they were selected for this study, and their significance in assessing cognitive performance.
- Clarify why a specific OR value (e.g., OR = 14.58 for the vigilance test) is considered significant or meaningful. Offer some interpretation of the effect size for the reader's understanding.
- If available, provide the statistical significance levels (p-values) for other variables in the negative binomial regression model. Readers may be interested in knowing if they were statistically significant even if they didn't exert a significant effect.
- If any figures, tables, or statistical analyses are referenced but not included in the provided text, ensure that these are added to the manuscript to support the findings discussed in the text.
- Consider including a brief conclusion or summary at the end of the results section to highlight the key findings and their implications.
Discussion and conclusion
Elaborate on the clinical implications of the variations in failure rates among different test variables. How might these variations impact the assessment of adult ADHD?
Discuss the implications of the differences in base rates between this study and previous research. Why do you think this study yielded higher rates, and what might be the potential reasons for this?
Address the limitations of the non-randomized test administration order. What potential biases or confounding factors might be introduced by the fixed order?
Provide suggestions for future research regarding test administration order and its potential effects on performance validity.
Offer more insights into the clinical implications of the high associations among EVIs. How can clinicians use this information to assess performance validity in practice?
Explore the implications of the limited correspondence between SVTs and PVTs. What might this reveal about the nature of symptom overreporting in the context of cognitive underperformance?
Discuss why SVTs and PVTs showed limited correspondence in this study. Are there specific clinical or demographic factors that might explain this observation?
Further clarify the need for future research to replicate the findings in different contexts, as well as the importance of known-group comparisons for validation.
Emphasize how addressing the limitations discussed might strengthen the validity of the results and their generalizability.
In the conclusions, consider summarizing the key takeaways from the study for clinicians and researchers. What are the most significant findings, and how should they be applied in practice?
Comments on the Quality of English Language
The topic is fascinating, but a thorough restructuring of the content is necessary. I kindly request the esteemed author to revise the entire article with a deeper and more coherent perspective, taking into consideration the suggested changes. Afterward, please resubmit it for further review. I believe this feedback will be valuable in improving the quality of your work.
Best regards.
Abstract
- Expand on the research objectives and clarify the specific research questions. For example, what is the significance of examining the order of test administration and its relationship with EVI failure rates? How does the prediction of cognitive underperformance by clinical information contribute to the study's goals?
- Mention the significance or potential implications of these objectives for the field of adult ADHD assessment.
- Provide more details about the neuropsychiatric sample. What are the characteristics of the participants? Mention any inclusion/exclusion criteria.
- Explain why the Vienna Test System (VTS; CFADHD) was chosen for the neuropsychological assessment and why it is suitable for this study.
- Describe the specific neuropsychological tests used for embedded performance validity indicators (PVTs). This will help readers understand the methodology better.
- Mention how the self- and other report symptom rating scales were chosen and why they are appropriate for assessing depressive symptoms and cognitive functioning.
- Elaborate on which specific PVTs or neuropsychological tests were the most sensitive for detecting underperformance. Provide relevant statistical information if available.
- Explain the significance of cognitive underperformance as a coherent construct and how it can influence the diagnostic process.
Introduction
- Provide a more concise and structured overview of the key points. Break down this section into shorter paragraphs for readability.
- Define "noncredible symptom report" and "noncredible performance" more explicitly.
- Discuss the potential impact of noncredible symptom reports and test performance on patients and the healthcare system.
- Cite the specific studies (references [1-3]) to give readers a clearer idea of the evidence in this area.
- Offer examples of specific EVIs derived from cognitive tests and their relevance to ADHD assessment.
- Clarify any variations in sample composition, methodology, or criteria used across the studies that may explain differences in base rates.
- Please for enrichment your introduction about ADHD, use the below references:
http://journalaim.com/Article/aim-20829
https://brieflands.com/articles/ijpbs-108390.html
- Address the exceptions or contradictions in the literature (e.g., White et al. 2022) in more detail.
- Explore the need for further research to compare the value of different embedded validity indicators (EVIs) across multiple cognitive domains.
- Consider discussing the challenges associated with assessing credibility in real-world clinical settings versus controlled studies.
- Clarify the role of the current study in building upon or validating the findings of Becke et al. (2023).
Materials and Methods
- Provide a more detailed description of the demographic characteristics of the participants, including age range, gender distribution, and education levels.
- Suggestion: Explain in more detail how the cognitive tests on the Vienna Test System were chosen and justify why they were considered appropriate for the study.
- Discuss the reliability and validity of the neuropsychological assessment tools and provide relevant references to support their use in this context.
- While conducting retrospective studies is acceptable, provide a more thorough explanation of the ethics protocol and the reasons for seeking ethical permission towards the end of data collection.
- Expand on the characteristics of participants, including their age range, gender distribution, and educational background. This will help readers better understand the diversity of the sample.
- Provide more context for the comorbid psychiatric disorders observed in the participants and their potential impact on the study.
- Include a brief statement about the ethical considerations for recruitment and data collection, explaining how informed consent was obtained.
- Mention the reliability and validity of the questionnaires used to assess ADHD symptoms and other relevant psychopathology.
Results
- Provide more context on what EVIs are, why they were selected for this study, and their significance in assessing cognitive performance.
- Clarify why a specific OR value (e.g., OR = 14.58 for the vigilance test) is considered significant or meaningful. Offer some interpretation of the effect size for the reader's understanding.
- If available, provide the statistical significance levels (p-values) for other variables in the negative binomial regression model. Readers may be interested in knowing if they were statistically significant even if they didn't exert a significant effect.
- If any figures, tables, or statistical analyses are referenced but not included in the provided text, ensure that these are added to the manuscript to support the findings discussed in the text.
- Consider including a brief conclusion or summary at the end of the results section to highlight the key findings and their implications.
Discussion and conclusion
Elaborate on the clinical implications of the variations in failure rates among different test variables. How might these variations impact the assessment of adult ADHD?
Discuss the implications of the differences in base rates between this study and previous research. Why do you think this study yielded higher rates, and what might be the potential reasons for this?
Address the limitations of the non-randomized test administration order. What potential biases or confounding factors might be introduced by the fixed order?
Provide suggestions for future research regarding test administration order and its potential effects on performance validity.
Offer more insights into the clinical implications of the high associations among EVIs. How can clinicians use this information to assess performance validity in practice?
Explore the implications of the limited correspondence between SVTs and PVTs. What might this reveal about the nature of symptom overreporting in the context of cognitive underperformance?
Discuss why SVTs and PVTs showed limited correspondence in this study. Are there specific clinical or demographic factors that might explain this observation?
Further clarify the need for future research to replicate the findings in different contexts, as well as the importance of known-group comparisons for validation.
Emphasize how addressing the limitations discussed might strengthen the validity of the results and their generalizability.
In the conclusions, consider summarizing the key takeaways from the study for clinicians and researchers. What are the most significant findings, and how should they be applied in practice?
